# Associations Between Body Appreciation, Body Weight, Lifestyle Factors and Subjective Health Among Bachelor Students in Lithuania and Poland: Cross-Sectional Study

**DOI:** 10.3390/nu16223939

**Published:** 2024-11-18

**Authors:** Vilma Kriaučionienė, Danuta Gajewska, Asta Raskilienė, Joanna Myszkowska-Ryciak, Julia Ponichter, Lina Paulauskienė, Janina Petkevičienė

**Affiliations:** 1Health Research Institute, Faculty of Public Health, Lithuanian University of Health Sciences, Tilžės Str. 18, 47181 Kaunas, Lithuania; asta.raskiliene@lsmu.lt (A.R.); janina.petkeviciene@lsmu.lt (J.P.); 2Department of Preventive Medicine, Faculty of Public Health, Lithuanian University of Health Sciences, Tilžės Str. 18, 47181 Kaunas, Lithuania; lina.paulauskiene@lsmu.lt; 3Department of Dietetics, Institute of Human Nutrition Sciences, Warsaw University of Life Sciences, 159C Nowoursynowska Str., 02-776 Warsaw, Poland; danuta_gajewska@sggw.edu.pl (D.G.); joanna_myszkowska-ryciak@sggw.edu.pl (J.M.-R.); julia_ponichter@sggw.edu.pl (J.P.); 4Institute of Biological Systems and Genetic Research, Lithuanian University of Health Sciences, Eivenių Str. 4, 44307 Kaunas, Lithuania

**Keywords:** students, body appreciation, body weight, nutrition habits, physical activity, subjective health

## Abstract

**Background/Objectives:** Positive body image is linked to improved mental and physical well-being, healthier lifestyles, and fewer unhealthy weight control behaviors. Cultural factors also play a role in influencing body appreciation. This study investigated the associations between body appreciation, body weight, lifestyle factors, and subjective health among bachelor’s students in Lithuania and Poland. **Methods:** A cross-sectional online survey was conducted with 1290 students from universities in both countries. The Body Appreciation Scale-2 (BAS-2) measured body appreciation, while participants provided self-reported data on their dietary habits, physical activity, sleep, health perceptions, and body weight and height. Linear regression models explored associations between BAS-2 scores, actual and perceived body weight, lifestyle habits, and subjective health. **Results:** Gender and country-based differences in body appreciation were observed. Lithuanian female students reported a higher median BAS score of 33 compared to 32 among Polish female students (*p* = 0.02), despite having a higher median BMI (22.3 kg/m^2^ vs. 21.1 kg/m^2^, *p* = 0.001). Positive body appreciation was linked to healthier dietary behaviors, such as higher consumption of fruits, vegetables, fish, and regular breakfasts. Additionally, greater physical activity and sufficient sleep were associated with higher body appreciation, while higher intake of sweets, sugary drinks, and fast food correlated with lower BAS-2 scores. Both BMI and perceived weight were negatively associated with body appreciation, particularly among females. **Conclusions:** Body appreciation is closely linked to body weight, healthier lifestyle, and positive health perceptions, suggesting that promoting healthier habits may improve body appreciation.

## 1. Introduction

Body appreciation is the recognition and acceptance of one’s body, regardless of its size, shape, or appearance. It means maintaining a positive attitude toward the body, focusing on its functionality and uniqueness rather than just its physical appearance or how it fits society’s beauty standards. Researchers have become increasingly interested in studying body appreciation to explore its associations with lifestyle, health perception, and overall well-being. The Body Appreciation Scale-2 (BAS-2), developed by Tylka and Wood-Barcalow, is a validated tool used to assess positive attitudes toward one’s body [1]. The study has demonstrated strong internal consistency (α = 0.97), test–retest reliability (r = 0.90), and construct validity (convergent, incremental, and discriminant) for the BAS-2 [1]. The scale has also been shown to be invariant across genders and was considered a reliable and valid tool for measuring the body appreciation of students [2].

Young adulthood, particularly in a university setting, is a critical period for identity formation, including self-image and health habits, which are often shaped by social, cultural, and academic environments [3]. University students face unique pressures related to academic performance, social integration, and, in recent years, the increased influence of social media on appearance ideals, which can significantly impact body appreciation and lifestyle behaviors. Additionally, university environments are structured to facilitate recruitment for large-scale surveys, as students are typically accessible through institutional channels, enabling the analysis of a relatively diverse sample across demographic groups. University students often come from varied backgrounds, including different regions, family structures, and socioeconomic statuses, offering a good possibility to analyze diverse populations [3].

Previous studies have demonstrated that higher BAS scores are associated with healthier behaviors. Instead of engaging in extreme dieting or harmful weight loss practices, individuals with high body appreciation were more likely to adopt a balanced and sustainable diet, along with moderate physical activity, respecting the body’s needs [1,4,5,6]. Research also shows that students with greater body appreciation are engaged in healthier behaviors, such as regular physical activity and healthier eating patterns. On the contrary, those with lower body appreciation scores often display poorer dietary choices and are more prone to sedentary lifestyles [2,5,7].

Gender differences in body appreciation have been well-documented. Studies have shown that women generally report lower BAS scores compared to men, largely due to societal emphasis on thinness for women and muscularity for men [8,9]. Body dissatisfaction among young women strongly predicts disordered eating, depressive symptoms, and unhealthy weight control practices. Conversely, men’s body image concerns focus more on muscularity and strength [10]. A study by Wawrzyniak et al. found that body dissatisfaction was more frequently reported among female adolescents and increased with age for both genders [11]. The literature describes the high prevalence of body weight dissatisfaction among female adolescents as “normative discontent” [12]. Furthermore, men are more likely to underestimate their body weight compared to women, even though a larger proportion of men are classified as overweight [13,14].

Social media’s influence on body appreciation has received considerable attention. Social platforms often promote narrow beauty ideals, leading to body dissatisfaction, particularly among women who engage in social comparisons. Studies have shown that increased use of social media intensifies body image concerns and negatively affects body appreciation, especially in cultures that highly value thinness and physical appearance [15,16].

Cultural differences play a significant role in body appreciation. While some studies indicate that body image concerns exist worldwide, the level of body appreciation can vary based on cultural and regional norms [6,17]. For instance, a study conducted across several European countries found that Polish students reported higher levels of body dissatisfaction compared to their peers in Lithuania. This discrepancy reflects differing societal norms regarding beauty and body size [18].

This study aims to examine the associations between body appreciation, body weight, and lifestyle factors among bachelor students in Lithuania and Poland. Although these two neighboring countries share some similarities, they also exhibit differences in dietary habits and societal norms, which may influence perceptions of body image

## 2. Materials and Methods

### 2.1. Study Design and Sample

The online cross-sectional study was conducted among bachelor students at the largest Universities of Applied Sciences in Lithuania, located in Vilnius, Kaunas, Klaipėda, and Šiauliai. Additionally, it included major Polish universities such as the Warsaw University of Life Sciences, Warsaw University of Technology, University of Gdańsk, and Poznań University of Economics. The survey at the Lithuanian universities took place during the second semester of the 2021–2022 academic year. The study in Poland was conducted from February to December 2022.

The data collection process involved sending emails to all students in the randomly selected faculties, providing information about the survey, and inviting them to participate. The online survey was open for three weeks, and additional reminders were sent during the first and second weeks to encourage completion. Participation in the study was voluntary and anonymous, ensuring that respondents felt comfortable providing their answers.

A self-administered questionnaire was completed by 1290 students: 709 students (216 males and 493 females) from Lithuanian universities, and 581 (121 males and 460 females) from Polish universities.

The study protocol received ethical approval from the Bioethics Centre at the Lithuanian University of Health Sciences (protocols BEC-GVM(M)-80, BEC-GM(M)-119) and the Ethics Committee for Research with Human Participation at the Institute of Human Nutrition Sciences of the Warsaw University of Life Sciences (Resolutions No. 3/2022 from 28 January 2022). Additionally, permission to conduct the study was obtained from the administration of the participating universities, ensuring compliance with institutional regulations and policies.

### 2.2. Measurements

A standardized questionnaire developed for this study was used in both countries. Students were asked about the frequency of consumption of selected foods, their physical activity levels, harmful behaviors, self-reported weight and height, attitudes toward weight gain, healthy eating, health perception, and body appreciation. A food frequency questionnaire was employed to assess the students’ dietary habits. Respondents were asked to indicate the frequency of eating breakfast and consuming 19 various food items, such as meat and meat products, poultry, fish/seafood, milk and dairy products, bread, cereal products, fresh vegetables, fruits, nuts/seeds, confectionery, sweets, soft drinks, energy drinks, fast foods, and snacks. Response options included the following categories: (1) ‘never’, (2) ‘1–4 times a month’, (3) ‘several times a week’, (4) ‘daily’, and (5) ‘several times a day’. Based on the reported frequency of food consumption, respondents were categorized into two groups: (1) ‘At least several times a week’ (‘several times per day’, ‘daily’ and ‘several times a week’) and (2) ‘1–4 times a month or never’.

Physical activity was assessed by asking two questions: (1) ‘How much time do you spend sitting on a typical weekday (e.g., sitting at a desk, watching television, reading)?’ and (2) ‘In your leisure time, how often do you engage in physical exercise for at least 30 min that makes you at least mildly short of breath or perspire?’ The possible answers were: ‘every day’, ‘4–6 times a week’, ‘2–3 times a week’, ‘once a week’, ‘once a month’, and ‘never’ coded from 1 to 7. Based on their responses, participants were categorized into two groups: (1) those physically active at least four times a week and (2) those physically active less often. Additionally, participants were asked to report their average daily sleep duration in hours.

Participants were also asked to report their actual weight (in kilograms) and height (in centimeters). Body mass index (BMI) was calculated by dividing weight (in kg) by the square of height (in meters). BMI values were categorized according to the WHO criteria: underweight (BMI < 18.5 kg/m^2^), normal weight (BMI 18.5–24.9 kg/m^2^), overweight (BMI 25–29.9 kg/m^2^), and obesity (BMI ≥ 30 kg/m^2^) [19]. Additionally, students were asked, ‘Would you be concerned if you gained weight?’, with possible five answers from (1) ‘never’ to (5) ‘very often. The question ‘How important is it for you to eat healthy?’ had five responses from (1) ‘very important’ to (5) ‘not at all’. Regarding health perception, students were questioned to what extent they keep an eye on their health. Response options included: ‘not at all’, ‘not much’, ‘to some extent’, and ‘very much’. Respondents were categorized as those who kept an eye on health answering ‘to some extent’ and ‘very much’ and the rest. Additionally, participants were asked about their health: ‘How would you assess your present health status?’, with the following answer categories: (1) ‘good’, (2) ‘reasonably good’, (3) ‘average, (4) ‘rather poor’, and (5) ‘poor’. Those who chose answers ‘good’ and ‘reasonably good’ were grouped as having a positive health assessment.

The Body Appreciation Scale-2 (BAS-2) was used to assess students’ body appreciation [1]. This scale includes 10 items, each rated on a five-point Likert scale ranging from 1 (never) to 5 (always). In the current study, Cronbach’s alpha coefficient was α = 0.962 for the Lithuanian version and 0.945 for the Polish version, indicating a strong internal consistency of the questionnaire. The overall score was determined by adding all responses, with a minimum sum of 10 and a maximum of 50. Higher scores reflect a more positive perception of body image.

### 2.3. Statistical Analysis

The categorical variables were presented as percentages and were compared using the Pearson chi-square test and z-test with Bonferroni correction for multiple comparisons. The continuous variables were presented as median and interquartile range, as all analyzed variables did not meet the criteria for normal distribution (Kolmogorov–Smirnov test). The Mann–Whitney test was used to compare the distributions of continuous variables between countries. The associations of BAS scores with BMI and perceived weight were analyzed using Spearman correlation analysis by country and gender. Multivariable linear regression analysis was used to associate the BAS scores with lifestyle factors, health perception, and other variables. Our data satisfied the assumptions of linear regression: (1) the response variable (BAS) was continuous, while the explanatory variables were either continuous or binary (gender and country); (2) the relationship between the outcome and the explanatory variables was linear; (3) the residuals were normally distributed; and (4) there was no multicollinearity among the explanatory variables. Initially, separate models were calculated for each variable, adjusting for gender, age, country, and BMI. Subsequently, variables that were uncorrelated and contributed to an increase in R square were included in the final model.

Data analysis was conducted using the IBM SPSS Statistics software package, version 29.0 (IBM Corp.: Armonk, NY, USA, released 2022). *p*-values < 0.05 were considered as significant.

## 3. Results

The main characteristics of the study population are presented in Table 1. The median age for men was 19 years in Lithuania and 20 years in Poland (*p* = 0.01), while for women, it was 20 years in both countries (*p* = 0.359).

Body appreciation levels were comparable for males in both Lithuania and Poland, with median BAS scores of 34 and 35, respectively. In contrast, Lithuanian female students had a higher median BAS score of 33, compared to 32 for Polish female students (*p* = 0.02). While the median BMI for males was similar in both countries, Lithuanian female students had a higher median BMI of 22.3 kg/m^2^, compared to 21.1 kg/m^2^ for their Polish counterparts (*p* = 0.001).

Polish students reported spending more time sitting than Lithuanian students, 8 h per day and 6 h, respectively. Additionally, Polish students consumed more fruits and vegetables than Lithuanians. They also had a higher frequency of breakfast consumption. Lithuanian students ate fish, legumes, nuts, and seeds more frequently than their Polish counterparts. Concerns about weight gain were more common among Polish students, whereas Lithuanian students tended to rate their health more positively than their Polish peers.

Most students in both countries were classified as having a normal weight (Figure 1). A higher proportion of males in Lithuania and Poland were categorized as overweight, while a smaller proportion were classified as underweight when compared to females. The distribution of males across body weight status categories was quite similar in both countries. In contrast, Polish female students showed a slightly higher prevalence of underweight (15.7%) and a lower prevalence of overweight (10.9%) compared to Lithuanian females, who had rates of 10.5% for underweight and 18.7% for overweight.

A larger percentage of Lithuanian males described themselves as ‘just right’ compared to Polish males, while more Polish males perceived themselves as ‘too fat’ (Figure 2). Among females, weight perceptions were more consistent between the two countries. Additionally, more females in both countries considered themselves ‘too thin’ compared to their male counterparts. In Lithuania, a higher proportion of females than males perceived themselves as ‘too fat’, while in Poland, more females than males considered themselves ‘just right’.

The Spearman correlation analysis indicated a significant negative association between BAS scores and BMI among female students in both Lithuania and Poland (Table 2). For male students, this negative association was statistically significant only in Poland. Additionally, perceived weight demonstrated a negative correlation with body appreciation for both genders in Poland and females in Lithuania, suggesting that students who perceive themselves as overweight tend to have lower BAS scores.

The results of the multivariable linear regression analysis, which calculated separate models for each variable, adjusting for gender, age, country, and BMI, indicated that body appreciation is associated with lifestyle, health perceptions, and other analyzed factors. (Table 3). A higher intake of vegetables and fruits, along with regular breakfast consumption, was positively associated with BAS scores. Additionally, higher frequencies of consuming fish, porridge, legumes, and nuts were linked to better body appreciation. On the contrary, a high intake of sweets, sugary drinks, fast food, and unhealthy snacks was related to lower BAS scores, suggesting that limiting less nutritious foods might contribute to a more positive body image. Students who considered healthy eating important had higher BAS scores compared to those with the opposite view.

Students who engaged more frequently in leisure-time physical activity reported higher body appreciation, while longer hours of sitting negatively affected BAS scores. Adequate sleep also played a significant role, as it was associated with better body appreciation, underscoring the importance of rest in promoting a positive body image.

Health perception was also associated with BAS. Students who rated their health more positively or took care of their health had higher BAS scores. However, those worried about potential weight gain tended to have lower BAS scores.

It is important to note that more statistically significant associations were observed among Lithuanian students compared to Polish students. In Poland, BAS scores were not statistically significantly related to the consumption of sweets, sugary drinks, fast food, snacks, or hours spent sitting, although trends were similar to those in Lithuania.

Multivariable linear regression analysis was conducted, including all uncorrelated variables that contributed to an increase in the R-squared value (Table 4). The results indicated that body appreciation was negatively correlated with several factors: BMI, the frequency of sweets consumption, a decrease in exercise frequency, a declining perception of health, concerns about weight gain, and being a Polish student. Conversely, body appreciation was positively associated with increased consumption of fish and nuts, longer sleep hours, and age. Additionally, males had higher BAS scores compared with females. Overall, this model accounted for 36.2% of the variations in BAS scores among students.

## 4. Discussion

This study examined body appreciation, BMI, lifestyle, and other factors among university students in Lithuania and Poland. Notable differences in the analyzed characteristics were observed between genders and countries. The results revealed strong associations between body appreciation and both actual and perceived body weight, as well as lifestyle choices, and subjective health.

Consistent with earlier studies, the dietary habits of Lithuanian and Polish students do not align with recommended dietary guidelines. A significant number of students consume insufficient amounts of vegetables, fruits, fish, legumes, and nuts, while frequently eating sweets, sugary drinks, fast food, and snacks. Other research has also shown that university students tend to have high-calorie diets that are rich in sugar and fat but low in essential nutrients, including fruits, vegetables, and fish [13,20,21,22]. Meal skipping, particularly breakfast, is common and has been linked to unhealthy patterns such as weight gain [20,22,23].

In our study, Lithuanian students reported consuming fish, legumes, nuts, and seeds more frequently than their Polish counterparts. Previous studies conducted in Poland, Germany, and Slovakia have also indicated differences in diet quality between these countries, highlighting that Polish students tend to consume a less healthy diet [18,24]. The highest level of knowledge about food and nutrition was observed among students from Poland. However, this knowledge did not correlate with a healthier diet [24]. Polish adolescents aged 13 to 19 who were dissatisfied with their body weight were less likely to meet dietary recommendations [11]. Many respondents expressed a willingness to engage in various activities aimed at increasing exercise, changing eating habits, or seeking assistance from professionals, such as personal trainers, to achieve their desired body shape. Conversely, concerns about gaining weight and losing attractiveness were commonly mentioned as factors influencing their perceptions of personal appearance [25].

It is important to note that dietary habits among Lithuanian students have improved from 2000 to 2017, with females exhibiting healthier eating patterns than males [13]. Moreover, Polish females also showed healthier food choices compared to males [11,26,27]. Other studies confirm that males are more likely to consume poorer diets, characterized by fast food and snacks, compared to females [13,28,29,30].

Our data indicate that the majority of students in both countries were of normal weight. Overweight was more prevalent among males, while underweight was more common in females. Polish females demonstrated slightly higher rates of underweight and lower rates of overweight compared to their Lithuanian counterparts. The COVID-19 pandemic contributed to weight gain among students, with some of these changes persisting even after the pandemic [31,32]. Overweight and obesity are increasing globally, affecting young people, including students, and influencing body image [33,34,35]. Students with higher BMI are more likely to experience body dissatisfaction and engage in unhealthy behaviors [31,36].

In our study, Polish females reported significantly lower BAS scores compared to Lithuanian females. This difference occurred despite Polish females having a lower prevalence of overweight. Additionally, Polish female students expressed more concern about weight gain than Lithuanian students. In terms of male perceptions, Polish students were more likely to believe they were overweight compared to Lithuanian male students, even though the prevalence of overweight and obesity was similar among males in both countries. These findings underscore the complex influence of cultural and gender-specific factors on body image perceptions in Lithuania and Poland [4,37].

Research data have shown that females are generally more dissatisfied with their weight and appearance and are more likely to engage in weight control practices compared to males [10,38]. In contrast, males often focus on muscularity, frequently viewing themselves as underweight or insufficiently muscular [39,40]. Social media has intensified body image concerns by promoting an ideal of thinness for women and muscularity for men, which contributes to body dissatisfaction and unhealthy behaviors [1,41,42]. Social platforms amplify these pressures through likes and comparisons, fostering disordered eating, especially in cultures that idealize thinness [43,44,45,46].

Our data align with previous studies showing that a higher BMI is associated with lower body appreciation, except for Lithuanian male students. Other studies also found that individuals with a higher BMI generally reported lower BAS scores, indicating greater body dissatisfaction [25,47]. This association was particularly pronounced among females for whom societal pressures and the promotion of thinness as the ideal body type by the media intensify body image concerns [10]. Males experience less pressure regarding body size and BMI compared to females [8]. This difference may explain why Lithuanian male students did not demonstrate a significant association between BMI and body appreciation.

In our study, BAS scores were negatively correlated with perceived weight in both countries and across genders. Other authors demonstrated that people who see themselves as overweight, regardless of their actual weight, tend to experience lower levels of body appreciation and higher levels of psychological distress [34]. Additionally, individuals with greater body dissatisfaction are more likely to engage in harmful weight control practices, such as restrictive dieting and excessive exercise, which can have negative effects on both their physical and mental well-being [20,46,48]. Research indicates that students feel pressure regarding their body appearance in fitness centers [47]. Thus, the increase in obesity prevalence among students, linked to physical inactivity and poor dietary habits, may heighten the risk of body dissatisfaction, particularly in societies where thinness is idealized [13,33].

Our data indicated that body appreciation is linked to the dietary habits of students. Higher consumption of vegetables, fruits, fish, porridge, legumes, nuts, and regular breakfast was associated with higher BAS scores. This is consistent with findings, which show that healthier diets are related to a positive body image [49,50]. Conversely, the intake of sweets, sugary drinks, fast food, and snacks of Lithuanian students was associated with lower BAS scores. Earlier studies have also shown that poor diets are related to negative body image and poor mental health [50,51].

The current study found a positive association between physical activity and body appreciation, which supports previous findings indicating that regular exercise enhances body image and self-esteem [7]. In contrast, sitting time was associated with lower body appreciation among Lithuanian students, reflecting research that shows inactivity negatively affects body image and overall health [52,53]. The increasing prevalence of sedentary behavior, driven by longer screen time, is a global concern.

High levels of weight importance during adolescence were found to be predictive of persistent dieting and disordered eating during young adulthood for both males and females [54].

Health perceptions demonstrated a strong association with body appreciation. Our findings indicated that students who rated their health positively or were more health-conscious reported higher BAS scores. This aligns with previous research suggesting a relationship between body image perception and better health outcomes [50].

This study has several limitations that should be acknowledged. Firstly, the cross-sectional design limits the ability to establish causal relationships between body appreciation and lifestyle factors such as diet, physical activity, and health perceptions. Secondly, the study relied on self-reported data for weight and height, which may introduce bias due to participants underreporting or overreporting these anthropometric measures, as well as their dietary habits and body weight perceptions. Social desirability bias may also affect responses, especially regarding body image and health behaviors. Additionally, the sample consisted only of bachelor students from various universities in Lithuania and Poland, which may limit the generalizability of the findings to larger populations or students in other countries. The lack of data on socioeconomic status and mental health, both of which could influence body appreciation and lifestyle behaviors, is another limitation that needs to be considered in future research. Lastly, the analysis did not include detailed measurements of social media use, which is known to significantly impact body image and should be explored further.

Despite these limitations, the study has several significant strengths. It examines a large and diverse sample from two countries, which allows for meaningful cross-cultural comparisons. This provides valuable insights into how regional and cultural differences may affect body appreciation and related lifestyle factors. The study employs the validated Body Appreciation Scale-2, ensuring accurate and consistent measurement of body image across different genders and cultures. Additionally, by including various health-related behaviors such as diet, physical activity, and sleep, along with subjective health assessments, the study offers a comprehensive overview of factors connected to body appreciation.

Future research should include longitudinal studies to explore the causal relationships between body appreciation and health behaviors over time. Expanding the sample to include students from different regions or countries could enhance the generalizability of the findings. Additionally, examining the roles of mental health, socioeconomic status, and the influence of social media on body appreciation would provide more comprehensive insights. Interventions aimed at promoting a positive body image, particularly among female students, could be developed to encourage healthier lifestyle choices and improve overall well-being.

## 5. Conclusions

This study emphasizes the strong links between body appreciation, lifestyle factors, such as diet and physical activity, and health perceptions among university students in Lithuania and Poland. Female students, especially in Poland, reported lower levels of body appreciation despite having lower rates of overweight. This suggests that cultural and gender-specific pressures may influence body image. Higher body appreciation was associated with healthier eating habits, increased physical activity, better sleep quality, and positive health perceptions. Conversely, sedentary behavior and a higher intake of sweets and fast food were linked to lower body appreciation. Promoting healthier lifestyle habits may support improved body appreciation among students.

## Figures and Tables

**Figure 1 nutrients-16-03939-f001:**
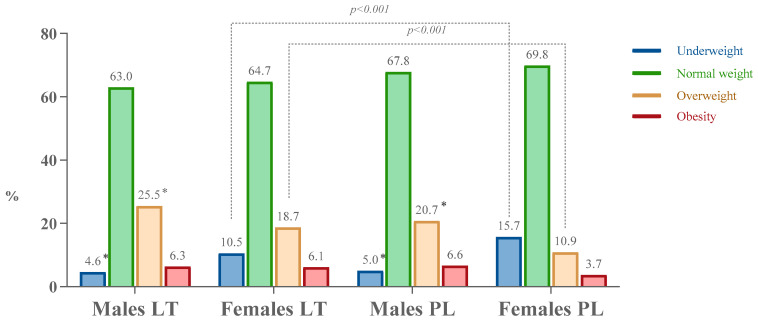
Distribution of male and female students by body weight status in Lithuania and Poland. * *p* < 0.05 compared to females in Lithuania or Poland (χ^2^ test with Bonferroni corrections); LT—Lithuania, PL—Poland.

**Figure 2 nutrients-16-03939-f002:**
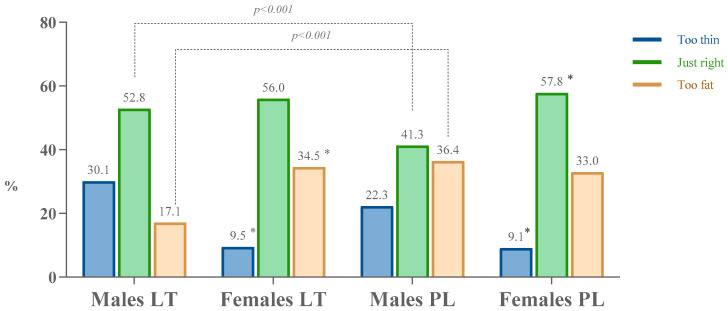
Distribution of male and female students by body weight perception in Lithuania and Poland. * *p* < 0.05 compared to males in Lithuania or Poland (χ^2^ test with Bonferroni corrections); LT—Lithuania, PL—Poland.

**Table 1 nutrients-16-03939-t001:** The characteristics of students in Lithuania and Poland.

Characteristics	Male Students			Female Students		
	Lithuania	Poland	*p*-Value	Lithuania	Poland	*p*-Value
Age (years), median (IR)	19 (2)	20 (3)	0.01	20 (4)	20 (3)	0.359
Body Appreciation Scale (scores), median (IR)	34 (10.8)	35 (13.5)	0.809	33 (12.5)	32 (14)	0.020
Body mass index, (kg/m^2^), median (IR)	23.3 (4.6)	23.5 (4.1)	0.834	22.3 (4.8)	21.1 (4.1)	0.001
Sitting time (hours), median (IR)	6 (4)	8 (4)	0.001	6 (4)	8 (4)	0.001
Sleep duration (hours)	7 (1)	7 (2)	0.001	7 (2)	7 (2)	0.294
Vegetable portions a day, median (IR)	1 (1)	2 (2)	<0.001	1 (1)	2 (2)	<0.001
Fruit portions a day, median (IR)	1 (1)	1 (2)	0.001	1 (1)	2 (1)	<0.001
Breakfast (days a week), median (IR)	5 (4)	7 (3)	0.025	5 (4)	7 (3)	<0.001
*Consumption of food products at least several times a week (% of participants)*						
Fish	43.1	14.9	<0.001	36.7	7.8	<0.001
Porridge/Cereals	62.5	47.9	0.010	58.6	62.6	0.208
Legumes	45.4	22.3	<0.001	42.0	33.0	<0.004
Nuts and seeds	65.3	32.2	<0.001	56.6	39.8	<0.001
Sweets (chocolate, candies)	60.2	56.2	0.476	56.6	65.0	0.008
Sugary drinks (soda)	53.2	52.1	0.836	35.1	40.7	0.077
Fast food	41.7	25.6	0.003	21.9	13.7	0.001
Snacks (chips, roasted peanuts, etc.)	51.4	42.1	0.103	31.2	27.6	0.220
*Perceptions and Behavior (% of participants)*						
Believe it is important to eat healthy	46.8	51.2	0.430	62.1	61.1	0.755
Worry about weight gain	7.4	15.4	<0.001	40.6	73.7	<0.001
Take care of health	77.8	65.3	0.013	86.4	62.8	<0.001
Evaluate health as good	81.5	62.0	<0.001	77.9	67.8	<0.001

IR—interquartile range; *p*-Value from Mann–Whitney or Pearson chi-square tests.

**Table 2 nutrients-16-03939-t002:** Spearman correlation coefficients between body appreciation scale and BMI as well as perceived weight.

Variable	Sex	Lithuania		Poland		Total	
		r	*p*-Value	r	*p*-Value	r	*p*-Value
BMI	M	−0.008	0.912	−0.284	0.002	−0.120	0.028
	F	−0.276	<0.001	−0.186	<0.001	−0.209	<0.001
Perceived weight	M	−0.133	0.051	−0.387	<0.001	−0.232	<0.001
	F	−0.386	<0.001	−0.362	<0.001	−0.372	<0.001

r—correlation coefficient; M—males; F—females.

**Table 3 nutrients-16-03939-t003:** Associations of body appreciation scale with lifestyle factors, health, and other variables (multivariable linear regression analysis *).

Variable	Lithuania			Poland			Total		
	β	CI	*p*-Value	β	CI	*p*-Value	β	CI	*p*-Value
Breakfast frequency	0.63	0.40; 0.86	<0.001	1.03	0.69; 1.38	<0.001	0.76	0.57; 0.96	<0.001
Vegetable daily portions	0.75	0.16; 1.34	0.013	0.61	0.25; 0.97	0.001	0.62	0.32; 0.92	<0.001
Fruit daily portions	0.73	0.13; 1.33	0.017	0.57	0.00; 1.13	0.049	0.64	0.24; 1.05	0.002
Fish consumption	1.65	0.86; 2.43	<0.001	1.711	0.46; 2.96	0.007	1.66	0.98; 2.33	<0.001
Porridge consumption	1.55	0.93; 2.17	<0.001	1.17	0.39; 1.96	0.003	1.32	0.84; 1.81	<0.001
Legumes consumption	1.68	0.97; 2.39	<0.001	1.11	0.17; 2.05	0.020	1.41	0.84; 1.98	<0.001
Nuts consumption	2.50	1.97; 3.03	<0.001	1.70	0.84; 2.56	<0.001	2.16	1.69; 2.62	<0.001
Sweets consumption	−1.83	−2.49; −1.18	<0.001	−0.41	−1.33; 0.51	0.381	−1.30	−1.84; −0.76	<0.001
Sugary drinks consumption	−1.58	−2.21; −0.96	<0.001	0.25	−0.47; 0.97	0.495	−0.68	−1.16; −0.21	0.005
Fast food consumption	−1.93	−2.91; −0.95	<0.001	−0.28	−1.76; 1.21	0.713	−1.36	−2.19; −0.53	0.001
Snacks consumption	−1.22	−2.05; −0.39	0.004	−0.40	1.50; 0.69	0.468	−0.89	−1.56; −0.22	0.009
Sitting hours	−0.38	−0.57; −0.18	<0.001	−0.28	−0.56; 0.01	0.057	−0.34	−0.50; −0.17	<0.001
Exercises during leisure time	−1.72	−2.07; −1.38	<0.001	−0.52	−0.97; −0.07	0.023	−1.18	−1.46; −0.89	<0.001
Sleep hours	1.65	1.18; 2.12	<0.001	1.34	0.70; 1.98	<0.001	1.49	1.11; 1.87	<0.001
Health perception	−1.70	−2.43; −0.97	<0.001	−4.43	−5.30; −3.56	<0.001	−2.87	−3.43; −2.30	<0.001
Taking care of health	3.34	2.48; 4.20	<0.001	3.67	2.76; 4.58	<0.001	3.52	2.90; 4.14	<0.001
Worrying about weight gain	−2.67	−3.18; −2.16	<0.001	−3.27	−3.80; −2.74	<0.001	2.97	2.60; 3.34	<0.001
Importance of eating healthily	−2.92	−3.58; −2.27	<0.001	−2.59	−3.35; −1.83	<0.001	−2.75	−3.24; −2.26	<0.001

* Separate models were calculated for each variable, adjusting for gender, age, and BMI, as well as the country in models where combined data were used. β—unstandardized regression coefficient; CI—confidence interval.

**Table 4 nutrients-16-03939-t004:** Associations of body appreciation scale with analyzed variables (multivariable linear regression analysis).

Variable	Unstandardized Coefficients	CI	*p*-Value	Standardized Coefficients
BMI	−0.22	−0.33; −0.11	<0.001	−0.10
Fish consumption	0.90	0.30; 1.50	0.003	0.08
Nuts consumption	1.36	0.93; 1.79	<0.001	0.16
Sweets consumption	−0.65	−1.12; −0.18	0.006	−0.06
Exercises during leisure time	−0.60	−0.85; −0.34	<0.001	−0.11
Sleep hours	0.92	0.59; 1.25	<0.001	0.13
Health perception	−2.08	−2.58; −1.58	<0.001	−0.19
Worrying about weight gain	−2.67	−3.00; −2.33	<0.001	−0.42
Age	0.23	0.13; 0.32	<0.001	0.11
Gender (males vs. females)	1.56	0.56; 2.56	0.002	0.08
Country (Poland vs. Lithuania)	−3.40	−4.31; −2.50	<0.001	−0.19

R square 0.362; CI—confidence interval.

## Data Availability

The data presented in this study are available on request from the corresponding author (This is due to ethical reasons as indicated in the Bioethics permissions).

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
