# Peer review of "Associations Between Body Appreciation, Body Weight, Lifestyle Factors and Subjective Health Among Bachelor Students in Lithuania and Poland: Cross-Sectional Study"

_nutrients, 2024, doi:10.3390/nu16223939_

Round 1
Reviewer 1 Report
Comments and Suggestions for Authors
1. add study design to the title
2. add significant differences in the abstract (numbers and p value)
3. please add explanation in the introduction why the study was conducted in students (why not in general population?)
4. Figure 1 could be improved, use for instance graph pad for figures, not power point
5. line 333 - why is the text striked out
6. conclusion should be shorter and according to the results
I believe this is a well conducted study on interesting topic such as body appreciation. I am not sure is it relevant in the field and how it adds to the body of literature in this field, maybe rationale for this could be added in the discussion. Tables and literature are appropriate.
Author Response
Comment 1: Add study design to the title
Response: The study design (cross-sectional study) was added to the title.
Comment 2: Add significant differences in the abstract (numbers and p value)
Response: The sentence with numbers and p value was added to the abstract: “Lithuanian female students reported a higher median BAS score of 33 compared to 32 among Polish female students (p = 0.02), despite having a higher median BMI (22.3 kg/m² vs. 21.1 kg/m², p = 0.001)”.
Comment 3: Please add explanation in the introduction why the study was conducted in students (why not in general population?)
Response: In response to the reviewer's suggestion, we expanded the introduction to clarify why the study focused on students: “Young adulthood, particularly in a university setting, is a critical period for identity formation, including self-image and health habits, which are often shaped by social, cultural, and academic environments [3]. University students face unique pressures related to academic performance, social integration, and, in recent years, the increased influence of social media on appearance ideals, which can significantly impact body appreciation and lifestyle behaviors. Additionally, university environments are structured to facilitate recruitment for large-scale surveys, as students are typically accessible through institutional channels, enabling the analysis of a relatively diverse sample across demographic groups. University students often come from varied backgrounds, including different regions, family structures, and socioeconomic statuses, offering a good possibility to analyze diverse populations [3].”
Comment 4: Figure 1 could be improved, use for instance graph pad for figures, not power point
Response: Figures 1 and 2 were improved using a graph pad.
Comment 5: Line 333 - why is the text striked out.
Response: Thank you for noticing the error. The strike-out words should have been deleted. We deleted them.
Comment 6: The conclusion should be shorter and according to the results
Response: The conclusion was modified as follows: “This study emphasizes the strong links between body appreciation, lifestyle factors, such as diet and physical activity, and health perceptions among university students in Lithuania and Poland. Female students, especially in Poland, reported lower levels of body appreciation despite having lower rates of overweight. This suggests that cultural and gender-specific pressures may influence body image. Higher body appreciation was associated with healthier eating habits, increased physical activity, better sleep quality, and positive health perceptions. Conversely, sedentary behavior and a higher intake of sweets and fast food were linked to lower body appreciation. Promoting healthier lifestyle habits may support improved body appreciation among students.”

Reviewer 2 Report
Comments and Suggestions for Authors
1. denote the Table 2 with “simple” linear regression analysis or “multiple“ linear regression analysis. If it was simple one, why did not the authors use correlation analysis?
2. For Table 3, please also denote below the Table with “multiple” linear regression analysis. Besides, please denote unstandardized or standardized regression coefficients, not just regression coefficient. Additionally, R squared value should also be reported. Furthermore, presenting both of unstandardized and standardized regression coefficients would be better. The latter can show relative importance of the predictors.
3. why did not the authors use all predictors (including Table 2 and Table 3) in predicting BAS scores? The two tables seemed that they could be combined. Or the BMI and perceived weight can be listed as predictors in Table 3. Besides, some significant characteristics of students between Lithuania and Poland in Table 1 can be considered into “Total” model in Table 3, such as age or the like.
4. The analytic logic was not consistent. Since the author reported median and IR, and used Mann-Whiney test, were they consistent with linear regression analysis based on continuous variable with normality assumption? Did the BAS scores meet normality? Besides, for Table 3, whether denoting that all the predictors were taken as binary variables would be better!
5.Some validity evidence of BAS-2 should be reported, not just cite some reference. It is better to show validity on basis of the present research data, such as showing factor loadings in the form of “0.65-0.83.” or the like.
6. the authors took the BAS as dependent variable in Table 2 and Table 3, but to write them in abstract “... Promoting body positivity among students may encourage healthier habits and improve overall well-being. Future research should focus on interventions aimed at fostering body appreciation in university settings.” Were BAS a predictor in this research? The causality should be cautious. The abstract and related wordings in conclusion section should be amended.
Author Response
Comment 1: Denote the Table 2 with “simple” linear regression analysis or “multiple“ linear regression analysis. If it was simple one, why did not the authors use correlation analysis?
Response: The simple linear regression analysis data was presented in Table 2. In response to the reviewer's suggestion, we replaced linear regression with correlation analysis, as shown in new Table 2. The description of the results was also changed.
Comment 2: For Table 3, please also denote below the Table with “multiple” linear regression analysis. Besides, please denote unstandardized or standardized regression coefficients, not just regression coefficient. Additionally, R squared value should also be reported. Furthermore, presenting both unstandardized and standardized regression coefficients would be better. The latter can show the relative importance of the predictors.
Response: Table 3 displays the results of the initial multivariable linear regression analysis, where separate models were calculated for each variable, controlling for gender, age, country, and BMI. This table includes only unstandardized regression coefficients. We provided an explanation in the Methods section and in the footnotes of Table 3.
Subsequently, variables that were uncorrelated and contributed to an increase in the R-squared value were included in the final model, which is presented in Table 4. In this final model, we display both unstandardized and standardized regression coefficients, along with the R-squared value.
Comment 3: Why did not the authors use all predictors (including Table 2 and Table 3) in predicting BAS scores? The two tables seemed that they could be combined. Or the BMI and perceived weight can be listed as predictors in Table 3. Besides, some significant characteristics of students between Lithuania and Poland in Table 1 can be considered into “Total” model in Table 3, such as age or the like.
Response: In the current version, Table 2 presents the results of the correlation analysis between BAS and BMI, and perceived body weight by gender and country. We aimed to highlight any gender differences in these associations. In the linear regression analyses shown in Table 3 and in the final model presented in Table 4, we included BMI, age, gender, and country as variables.
Comment 4: The analytic logic was not consistent. Since the author reported median and IR, and used Mann-Whiney test, were they consistent with linear regression analysis based on continuous variable with normality assumption? Did the BAS scores meet normality? Besides, for Table 3, whether denoting that all the predictors were taken as binary variables would be better!
Response: We can confirm that our data satisfied the assumptions of linear regression. We explained it in the Methods section: “Our data satisfied the assumptions of linear regression: (1) the response variable (BAS) was continuous, while the explanatory variables were either continuous or binary (gender and country); (2) the relationship between the outcome and the explanatory variables was linear; (3) the residuals were normally distributed; and (4) there was no multicollinearity among the explanatory variables.” All explanatory variables were continuous, including BMI, food consumption and physical activity frequencies, as well as scores of health perception and worries about weight gain. The only exceptions were gender and country, which were binary variables.
Comment 5: Some validity evidence of BAS-2 should be reported, not just cite some reference. It is better to show validity on basis of the present research data, such as showing factor loadings in the form of “0.65-0.83.” or the like.
Response: The Cronbach’s alpha coefficient was reported in the Methods section for the Lithuanian and Polish BAS-2 versions separately: “In the current study, Cronbach's alpha coefficient was α = 0.962 for the Lithuanian version and 0.945 for the Polish version, indicating a strong internal consistency of the questionnaire.”
Comment 6: The authors took the BAS as dependent variable in Table 2 and Table 3, but to write them in abstract “... Promoting body positivity among students may encourage healthier habits and improve overall well-being. Future research should focus on interventions aimed at fostering body appreciation in university settings.” Were BAS a predictor in this research? The causality should be cautious. The abstract and related wordings in conclusion section should be amended.
Response: The conclusion in the abstract was modified as follows: “In conclusion, body appreciation is closely linked to body weight, healthier lifestyle, and positive health perceptions, suggesting that promoting healthier habits may improve body appreciation.” In response to both reviewers’ suggestions, the final conclusion was also modified.
